# TEST-TIME ADAPTATION WITH CLIP REWARD FOR ZERO-SHOT GENERALIZATION IN VISION-LANGUAGE MODELS

**Shuai Zhao**[†,§]* **Xiaohan Wang**[‡] **Linchao Zhu**[‡] **Yi Yang**[‡]

[†]ReLER Lab, AAII, University of Technology Sydney
[‡]ReLER Lab, CCAI, Zhejiang University  [§]Baidu Inc.
{zhaoshuaimcc, wxh1996111}@gmail.com  {zhulinchao, yangyics}@zju.edu.cn

## ABSTRACT

One fascinating aspect of pre-trained vision-language models (VLMs) learning under language supervision is their impressive zero-shot generalization capability. However, this ability is hindered by distribution shifts between the training and testing data. Previous test time adaptation (TTA) methods for VLMs in zero-shot classification rely on minimizing the entropy of model outputs, tending to be stuck in incorrect model predictions. In this work, we propose TTA with feedback to rectify the model output and prevent the model from becoming blindly confident. Specifically, a CLIP model is adopted as the reward model during TTA and provides feedback for the VLM. Given a single test sample, the VLM is forced to maximize the CLIP reward between the input and sampled results from the VLM output distribution. The proposed *reinforcement learning with CLIP feedback (RLCF)* framework is highly flexible and universal. Beyond the classification task, with task-specific sampling strategies and a proper reward baseline choice, RLCF can be easily extended to not only discrimination tasks like retrieval but also generalization tasks like image captioning, improving the zero-shot generalization capacity of VLMs. According to the characteristics of these VL tasks, we build different fully TTA pipelines with RLCF to improve the zero-shot generalization ability of various VLMs. Extensive experiments along with promising empirical results demonstrate the effectiveness of RLCF. The code is available at https://github.com/mzhaoshuai/RLCF.

## 1 INTRODUCTION

Pre-trained vision-language models (VLMs) learning under language supervision (Radford et al., 2021; Jia et al., 2021; Yuan et al., 2021) exhibit promising zero-shot transferability. This encourages researchers to explore the capabilities of VLMs across a number of tasks in a zero-shot fashion. For example, Hong et al. (2022) employ CLIP for zero-shot text-driven avatar generation, Sain et al. (2023) adapt CLIP for zero-shot sketch-based image retrieval, and Li et al. (2023) achieve zero-shot image captioning without images. Nonetheless, the large domain gap between training and test data is still challenging for VLMs in a zero-shot circumstance. In this work, we investigate how to fulfill the domain gap during test time in various tasks without task-specific training corpus, namely, test time adaptation (TTA) for VLMs with a zero-shot prerequisite.

One pioneer TTA work in improving the zero-shot classification ability of VLMs is test time prompt tuning (TPT) (Manli et al., 2022). Given a single test sample, TPT optimizes the learnable prefix tokens by minimizing the entropy of model outputs to bootstrap its generalization capacity. Nevertheless, making the model confident in its predictions is a double-edged sword. It does reduce the test error and close the domain gap at a certain level (Wang et al., 2021a), but it makes the model stick to its incorrect predictions and unable to get out of the dilemma by itself as shown in the top of Figure 1a. Entropy minimization tends to make the model blindly confident.

---

*Part of this work is done during an internship at Baidu Inc. Yi Yang is the corresponding author.

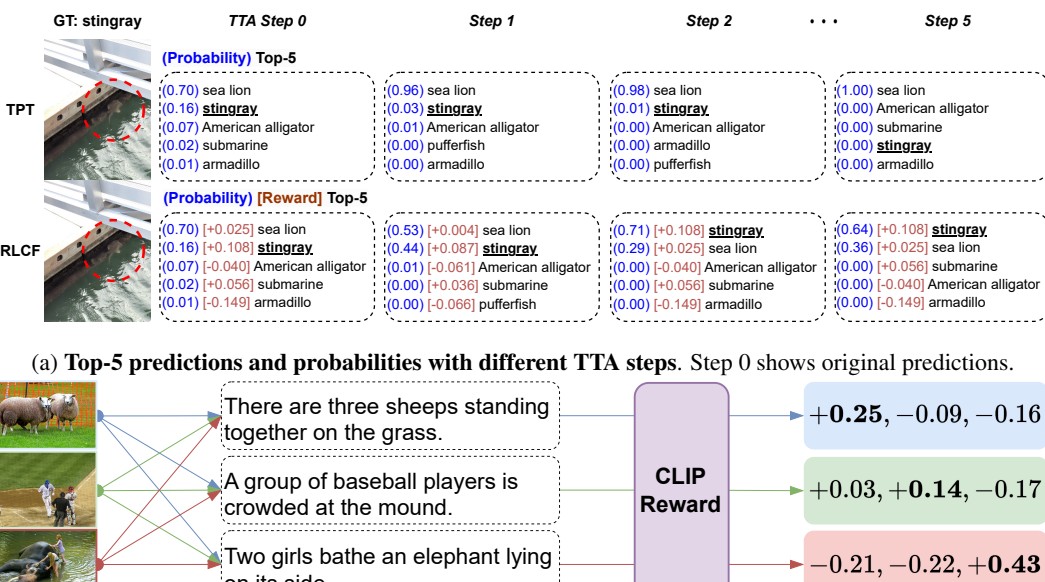

(a) **Top-5 predictions and probabilities with different TTA steps**. Step 0 shows original predictions.

(b) **Examples of CLIP reward**. The average score of an image and all sentences is the reward baseline.

Figure 1: **Feedback mechanism in zero-shot generalization with CLIP as the reward model**.

Inspired by the success of the feedback mechanism in language and vision tasks (Ouyang et al., 2022; OpenAI, 2023; Pinto et al., 2023), we introduce feedback during test time to rectify the VLM output as shown in the bottom of Figure 1a. Previous feedback methods leverage labeled preference data to train a reward model (Ouyang et al., 2022; Lee et al., 2023) or use labels to calculate the reward (Cho et al., 2022; Pinto et al., 2023). Without ground truth, we refer to the well-recognized CLIP (Radford et al., 2021) model as the feedback resource. CLIP shows powerful generalization capacity across many VL tasks. The outputs of CLIP are also well-calibrated (without fine-tuning on a specific dataset) (Minderer et al., 2021), *i.e.*, the score from CLIP accurately reflects its uncertainty about the input sample. This makes CLIP a reliable reward model. One more question is why feedback rather than directly tuning with CLIP supervision? Ouyang et al. (2022) demonstrate that model learning with feedback has better generalization abilities than a supervised fine-tuning model. We get the same conclusion from our empirical results. Furthermore, CLIP supervision cannot be directly used in generation tasks like image captioning, while the feedback mechanism is versatile.

Our proposed framework, coined as *reinforcement learning with CLIP feedback (RLCF)*, is flexible and universal for TTA with different VLMs in various tasks. With task-specific sampling strategies and a proper reward baseline choice, RLCF is applicable across zero-shot classification, text-image retrieval, and image captioning. In these tasks, the model is given a single test sample, we then sample $K$ candidates from the output distribution. For discrimination tasks like classification and retrieval, the top-$K$ sampling is applied; for the caption generalization task, a beam search method is adopted. Assuming the input is an image, like Figure 1b, the CLIP model first gives the CLIP-Score (Hessel et al., 2021) between the image and all candidate sentences. As CLIPScore is always non-negative, the average score is subtracted from the calculated scores. This average baseline aims to distinguish which model behaviors are encouraged and which are discouraged. Then the learnable parameters in the TTA model are optimized by REINFORCE (Williams, 1992) algorithm.

While the reward design and learning algorithm remain consistent across various tasks, the TTA pipelines are tailored to each specific task. For classification, we inherit the data augmentation and confidence selection pipeline from TPT (Manli et al., 2022), making it work for not only prefix tuning but also backbone adaptation. For retrieval, considering a large number of candidate entries, we only update the parameters *w.r.t.* the query for efficiency. For instance, we only tune the branch *w.r.t.* the input modality for a two-branch VLM like CLIP. For image captioning, we construct the TTA pipeline with two methods (Mokady et al., 2021; Nukrai et al., 2022) built upon large language models (LLMs). During TTA, we only tune the projector which projects the image into the LLM token embedding space. Plus, several task-agnostic practical tricks are applied, *i.e.*, multiple reward models, episodic TTA (Wang et al., 2021a), and momentum buffer for incremental learning.

To summarize our contributions: 1) To the best of our knowledge, RLCF is the first universal fully TTA framework for improving the zero-shot generalization capacity of VLMs across different tasks. 2) We develop a novel reward function for test time RL with CLIP. It is simple yet effective. Compared to previous methods (Cho et al., 2022) in the training stage, it demonstrates that CLIP can be used as a practical reward model alone, even with a single test sample. 3) We design task-specific TTA pipelines for three VL tasks with RLCF. Extensive experiments with promising results validate the effectiveness of RLCF in boosting the zero-shot performance of different VLMs.

## 2  RELATED WORK

**Reinforcement learning in language and vision**  The most well-known application of RL in natural language process is reinforcement learning with human feedback (RLHF) (Ouyang et al., 2022; OpenAI, 2023). A reward model is trained with preference data collected from humans, and it is used to fine-tune the LLM via proximal policy optimization (PPO, Schulman et al. (2017)). Similar approaches are applied in (Ziegler et al., 2019; Stiennon et al., 2020; Bai et al., 2022; Glaese et al., 2022). In prompt engineering for language models, RLPrompt (Deng et al., 2022) and TEMPERA (Zhang et al., 2023) search for discrete text prompts by RL. RL has also been widely applied in vision and multi-modal research. A comprehensive study of deep RL in computer vision can be found at (Le et al., 2022). Recently, Pinto *et al.* (Pinto et al., 2023) optimize vision task metrics using RL and achieve promising results, demonstrating the effectiveness of RL in vision. In the multi-modal area, ImageReward (Xu et al., 2023) collects preference data and trains a reward model for text-to-image generation tasks, similar to RLHF in GPT models. SCST (Rennie et al., 2017) apply CIDEr metric as a reward function with REINFORCE (Williams, 1992) algorithm to improve the generation quality in image captioning during training. Cho *et al.* (Cho et al., 2022) explore the possibility of using CLIPScore (Hessel et al., 2021) as the reward in image captioning. Their empirical results show that CLIPScore cannot be an independent reward function and should be combined with a grammar regularization or CIDEr metric. Nevertheless, the training of the grammar head and calculation of CIDEr metric both need reference text, which is unavailable at test time.

**Test-time adaptation**  Test-time adaptation (TTA) aims to address the distribution shift between training and test data during test time (Sun et al., 2020; Liu et al., 2021; Wang et al., 2021a). Test-time training (TTT, Sun et al. (2020)) allows modifications to the training pipeline. In such cases, self-supervised auxiliary tasks are incorporated to help the model adapt to the distribution of test data (Sun et al., 2020; Liu et al., 2021; Lin et al., 2023). For example, TTT+ (Liu et al., 2021) utilizes instance discrimination tasks from contrastive learning (Chen et al., 2020). On the other hand, fully TTA assumes that the training pipeline cannot be modified as the training data is unavailable (Wang et al., 2021a). Two popular techniques in fully TTA are normalization layer adaptation and entropy minimization. Normalization layer adaptation updates data statistics or parameters of the normalization layer based on batched test samples (Wang et al., 2021a; Schneider et al., 2020; Niu et al., 2023) or augmented data views from a single test sample (Zhang et al., 2022a). Entropy minimization aims to make the model confident in its predictions to reduce generalization error (Wang et al., 2021a; Zhang et al., 2022a; Manli et al., 2022; Niu et al., 2022; 2023). There is also a retrieval-augmented TTA method (Zancato et al., 2023), which uses CLIP to retrieve informative data from an external large dataset and update the decision boundary during test time.

## 3  METHOD

### 3.1  PRELIMINARIES

**Fully test-time adaptation in vision-language tasks**  Let $f_\theta(\cdot)$ represent a VLM trained on image-text pairs $\mathcal{D}_{train} = \{(t_i, v_i)\}_{i=1}^N$ with parameter $\theta$, where $t_i \in \mathcal{T}_{train}$ (training text space) and $v_i \in \mathcal{V}_{train}$ (training image space). The objective of TTA (Sun et al., 2020; Wang et al., 2021a) is to boost $f_\theta(t)$ or $f_\theta(v)$ on domain-shifted test samples $\mathcal{D}_{test} = \{(t_j)\}_{j=1}^M$ or $\mathcal{D}_{test} = \{(v_j)\}_{j=1}^M$, where $t_j \in \mathcal{T}_{test}$ (testing text space), $v_j \in \mathcal{V}_{test}$ (testing image space), $\mathcal{T}_{test} \neq \mathcal{T}_{train}$, and $\mathcal{V}_{test} \neq \mathcal{V}_{train}$. We assume that the VLM takes either text or image as input and outputs the other modality. In fully TTA, the training data are unavailable, and the training pipeline cannot be modified. Following TPT (Manli et al., 2022) and MEMO (Zhang et al., 2022a), the adaptation is conducted with a single test point, *i.e.*, the VLM is exposed to only one $t_j$ or $v_j$.

**Contrastive Language-Image Pre-training (CLIP)** CLIP (Radford et al., 2021) comprises an image encoder $g(\cdot)$ and a text encoder $h(\cdot)$. CLIP is pre-trained using a contrastive loss that encourages similarity between feature vectors of paired images and text, aligning them in a shared embedding space. Once pre-trained, CLIP can assess the similarity between the text $t$ and image $v$ as follows:

$$\texttt{CLIP}(t, v) = \cos(h(t), g(v)), \tag{1}$$

where $\cos(\cdot, \cdot)$ represents the cosine similarity. For image classification with CLIP, the input text consists of the prompt plus the class names, *i.e.*, $t = \{p_t; \text{"dog"}\}$, where prompt $p_t = \text{"a photo of a"}$.

### 3.2 TEST-TIME ADAPTATION WITH CLIP REWARD

#### 3.2.1 REINFORCEMENT LEARNING WITH CLIP FEEDBACK

Without loss of generality, we first consider the case where the VLM $f_\theta(\cdot)$ takes an image $v$ as input and maps it to text $t$. During TTA, our goal is to learn a conditional distribution $P(t|v, \theta) = f_\theta(v)$ that maximizes a reward function $\mathcal{R}(\cdot, \cdot)$. Formally, the optimization problem during TTA is:

$$\max_\theta \mathbb{E}_{t \sim P(\cdot|v, \theta)} \mathcal{R}(t, v). \tag{2}$$

Different from previous methods (Rennie et al., 2017; Cho et al., 2022; Pinto et al., 2023) which maximizes the expected reward over batched training samples, here we only maximize the expected reward over *a single test sample* $v \in \mathcal{V}_{test}$.

**Policy gradient with REINFORCE** To compute the gradient of the non-differentiable reward function, REINFORCE (Williams, 1992) is adopted to calculate $\nabla_\theta \mathbb{E}_{t \sim P}[\mathcal{R}(t, v)]$. It uses the so-called "log-derivative trick" to estimate the gradient of the expected reward for a given input:

$$\nabla_\theta \mathbb{E}_{t \sim P}[\mathcal{R}(t, v)] = \mathbb{E}_{t \sim P}[\mathcal{R}(t, v) \nabla_\theta \log P(t|v; \theta)]. \tag{3}$$

In a VL task, the input and output modalities are closely related, *e.g.*, the input is an image and the output is the description of the image. Therefore, we can use CLIP to evaluate the similarity between the input and output, and the model can maximize this similarity to align with task goals. Similar to Cho et al. (2022), we use CLIPScore (Hessel et al., 2021) as the reward:

$$\texttt{CLIP-S}(t, v) = w \times \max(\texttt{CLIP}(t, v), 0), \tag{4}$$

where $w = 2.5$ is a constant. CLIPScore is always non-negative, which means it encourages all model behaviors. However, for an irrelevant sampled image-text pair in Figure 1b, we expect the reward model to provide negative feedback to discourage such behavior. Cho *et al.* (Cho et al., 2022) adopt a greedy search baseline which needs to be combined with a grammar regularization or CIDEr metric to be a practical reward function. In this work, we demonstrate that with proper sampling strategies and baseline, CLIPScore can also be used as the sole reward function in different VL tasks. Specifically, we set the reward baseline as the average CLIPScore of sampled image-text pairs. The reward function with baseline becomes:

$$\mathcal{R}(t, v) = \texttt{CLIP-S}(t, v) - \mathbb{E}_{t \sim P}[\texttt{CLIP-S}(t, v)]. \tag{5}$$

It is straightforward to get the reward function for a VLM which takes text $t$ as input and return an image $v$ according to Eq. (5). The sampling strategies will be presented in the next section.

#### 3.2.2 TASK-SPECIFIC FULLY TEST-TIME ADAPTATION

RLCF is flexible and applicable across various VL tasks, and we apply RLCF to three different VL tasks in this work. For all tasks, the VLM solely learns through REINFORCE with Eq. (5) as the reward function during test time. However, VLMs and sampling strategies vary with tasks. Next, we introduce our task-specific fully TTA pipelines.

**Zero-shot image classification on OOD data** Figure 2 illustrates the fully TTA pipeline for zero-shot image classification with RLCF. Without loss of generality, we also choose CLIP as the classifier. The TTA pipelines include two adaptation manners: prompt tuning and image encoder tuning. TPT (Manli et al., 2022) shows that entropy minimization for image encoder tuning results in inferior performance compared to prompt tuning. By contrast, RLCF works both with prompt tuning and image encoder tuning, demonstrating its versatility.

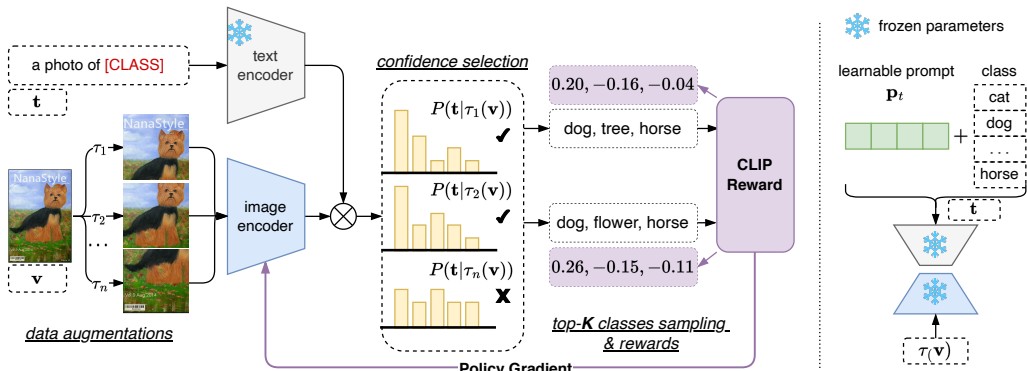

Figure 2: **Fully TTA for zero-shot image classification with CLIP reward.** *Left*: image encoder tuning. *Right*: prompt tuning. The pipelines of the two are the same except for the learnable parameters. A single test image is first augmented to produce multiple views, then only confident views with low-entropy predictions are selected. For each selected view, we sample the top-$K$ classes, calculate their rewards, and update the parameters using policy gradient.

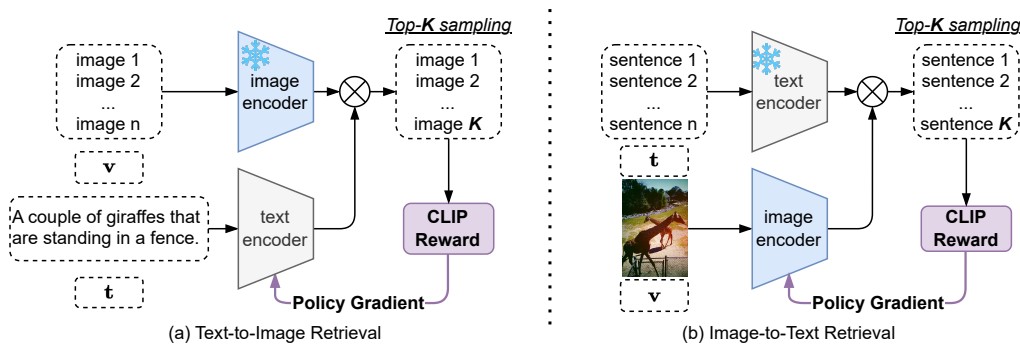

Figure 3: **Fully TTA for zero-shot text-image retrieval with CLIP reward**.

In Figure 2, given a test image $v$, it is first operated with data augmentors $\{\tau_1, \tau_2, \ldots, \tau_n\}$ for multiple different views. Following TPT (Manli et al., 2022) and SAR (Niu et al., 2023), we only reserve the confident samples with low-entropy predictions, namely, the entropy $H(P(t|\tau(v)))$ of the selected view should be low. High-entropy predictions are considered unreliable as they lack confidence in their outputs. In practice, we use the bottom 10th percentile of $n = 64$ augmented views with low entropies as TPT (Manli et al., 2022). For each low-entropy view, class names of the top-$K$ predictions are used to calculate their CLIP rewards according to Eq. (5). The learnable parameters are then optimized to maximize the expected reward by gradient descent as Eq. (3).

One point that needs to be clarified is why using the CLIP reward as feedback rather than directly fine-tuning the model with CLIP supervision. For example, methods like knowledge distillation (KD, Hinton et al. (2015)) or pseudo-label (Lee et al., 2013). InstructGPT (Ouyang et al., 2022) demonstrates that model learning with feedback has better generalization capabilities compared to a supervised fine-tuning model. In our context, KD or pseudo-label requires a weak model (student) to mimic a strong model (teacher). However, it is worth noting that the student may be correct while the teacher may be incorrect. For instance, given an image of a dog, the top-3 predictions of the student and teacher models are {dog, horse, tree} and {cat, dog, horse}, respectively. For KD or pseudo-label, the student will be forced to follow the incorrect behaviors of the teacher. In contrast, the feedback mechanism only assesses the sampled results from the student, less likely to alter the correct prediction. In such cases, the feedback mechanism combines the merits of both the student and the teacher. Another important reason is that CLIP supervision cannot be directly used in generalization tasks like image captioning, while the feedback mechanism is universal.

**Zero-shot text-image retrieval** The fully TTA pipeline for zero-shot retrieval with RLCF is presented in Figure 3. CLIP also serves as the zero-shot retrieval model. For retrieval, the number of candidates is usually large, so we only update the parameters with respect to the query for efficiency.

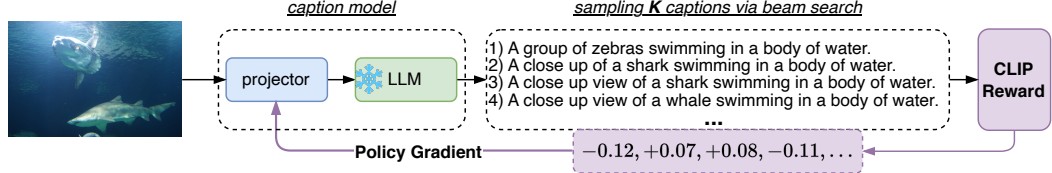

Figure 4: **Fully TTA for image captioning with CLIP reward.**

For text-to-image retrieval, the image encoder remains fixed, while the text encoder is frozen in the other case. Given a query, top-$K$ sampling is employed to the returned results to calculate the reward. Unlike image classification, no augmentations are used for the input query. The retrieval task requires a holistic understanding of the input query rather than identifying a single object. Augmentations like crop and flip may lead to corrupt semantics.

**Zero-shot and cross-domain image captioning** Figure 4 illustrates the fully TTA pipeline for image captioning with RLCF. The captioning TTA pipeline is built upon two LLM-based methods: CapDec (Nukrai et al., 2022) and CLIPCap (Mokady et al., 2021). CapDec is trained only with text and CLIPCap is trained with images. TTA with CapDec is undertaken with a zero-shot prerequisite, and TTA with CLIPCap is cross-domain. During the test, CapDec and CLIPCap will be given unseen and domain-shifted images, respectively. Both CapDec and CLIPCap utilize a projector (*e.g.*, an MLP or transformer (Vaswani et al., 2017)) to project CLIP feature vectors into the token embedding space of the LLM. Only the projector is updated through policy gradient, while the LLM remains fixed during TTA. Beam search is employed to sample $K$ generated captions for reward calculation.

### 3.2.3 TEST-TIME ADAPTATION TRICKS

In this section, we introduce several general TTA techniques applicable across different tasks.

**Multiple reward models with weights** By default, a single CLIP-ViT-L/14 is used as the reward model. An ensemble of multiple reward models can be used for better feedback. We assign scores based on human preference for different CLIP models: {CLIP-ViT-L/14-336: 10, CLIP-ViT-L/14: 5, CLIP-RN50×64: 3}. These scores are then normalized to sum up to 1, serving as weights for the ensemble. CLIP-RN uses a ResNet (He et al., 2016) as the image encoder, while CLIP-ViT adopts a vison transformer (Dosovitskiy et al., 2021).

**Episodic TTA** The model is exposed to only a single test sample once, making the learned knowledge unreliable for other samples. Hence, after each TTA process, the model parameters $\theta$ are reset to the initial state $\theta^\star$ like (Wang et al., 2021a; Manli et al., 2022). It is called episodic TTA.

**Momentum buffer** While episodic TTA ensures reliability, it limits the incremental learning ability of the model. To address this issue, we introduce a momentum buffer $\xi$, initialized as $\xi \leftarrow \theta^\star$. After a TTA process, $\theta$ becomes $\overline{\theta}$, and $\xi$ is updated by $\xi \leftarrow m\xi + (1 - m)\overline{\theta}$, where $m \in [0, 1)$ is a momemtum coefficient. Every $B_s$ samples, we update $\theta^\star \leftarrow \xi$. At the start of the next TTA process, $\theta \leftarrow \theta^\star$, allowing the utilization of the learned knowledge. The momentum buffer functions similarly to an ensemble of different models, resembling model soups (Wortsman et al., 2022).

## 4 EXPERIMENTS

This section presents the experimental TTA results in three tasks. For variants of our method, **RLCF** uses a CLIP-ViT-L/14 as the reward model, **RLCF-S** adopts weighted reward sum of {CLIP-ViT-L/14-336, CLIP-ViT-L/14, CLIP-RN50×64}, and **RLCF-S-M** adds the momentum buffer.

### 4.1 ZERO-SHOT IMAGE CLASSIFICATION ON OOD DATA

**Datasets** Following CLIP and TPT, we test RLCF on ImageNet (Deng et al., 2009) and its four variant test sets with distribution shifts: ImageNet-A (Hendrycks et al., 2021b), ImageNet-V2 (Recht et al., 2019), ImageNet-R (Hendrycks et al., 2021a), and ImageNet-Sketch (Wang et al., 2019). ImageNet-A consists 7,500 natural adversarial images misclassified by a ResNet-50. ImageNet-V2 contains 10,000 natural images from different sources. ImageNet-R collects 30,000 images with artistic renditions. ImageNet-Sketch includes 50,000 black and white sketch images.

Table 1: **Top-1 accuracy of zero-shot image classification with TTA on OOD data**. KD uses a CLIP-ViT-L/14 as the teacher. The best and second-best results are highlighted. Improvement in accuracy of RLCF compared to the baselines (zero-shot CLIP-ViT-B/16 or CoOp) is in (↑blue).

| Method | ImageNet | ImageNet-A | ImageNet-V2 | ImageNet-R | ImageNet-Sketch | OOD Average |
|---|---|---|---|---|---|---|
| | | | *Zero-shot baseline* | | | |
| CLIP-ViT-B/16 | 66.73 | 47.87 | 60.86 | 73.98 | 46.09 | 57.20 |
| CLIP-ViT-L/14 | 73.44 | 68.82 | 67.80 | 85.40 | 57.84 | 69.97 |
| | | | *Prompt tuning for* CLIP-ViT-B/16 | | | |
| CoOp (Zhou et al., 2021) | 71.51 | 49.71 | 64.20 | 75.21 | 47.99 | 59.28 |
| CoCoOp (Zhou et al., 2022) | 71.02 | 50.63 | 64.07 | 76.18 | 48.75 | 59.91 |
| TPT (Manli et al., 2022) | 68.98 | 54.77 | 63.45 | 77.06 | 47.94 | 60.81 |
| TPT + CoOp (Manli et al.) | 73.61 | 57.95 | 66.83 | 77.27 | 49.29 | 62.84 |
| TPT + CoOp + KD (Hinton et al.) | 71.40 | 63.25 | 65.28 | 82.70 | 55.78 | 66.75 |
| **RLCF** | 73.23$_{(↑6.50)}$ | 65.45$_{(↑17.58)}$ | 69.77$_{(↑8.91)}$ | 83.35$_{(↑9.37)}$ | 54.74$_{(↑8.65)}$ | 68.33$_{(↑11.13)}$ |
| **RLCF + CoOp** | 76.05$_{(↑4.54)}$ | 69.74$_{(↑20.03)}$ | 70.62$_{(↑6.42)}$ | 84.51$_{(↑9.30)}$ | 56.49$_{(↑8.50)}$ | 70.34$_{(↑11.06)}$ |
| **RLCF-S + CoOp** | 76.50$_{(↑4.99)}$ | 71.11$_{(↑21.40)}$ | 70.92$_{(↑6.72)}$ | 84.73$_{(↑9.52)}$ | 56.97$_{(↑8.98)}$ | 70.93$_{(↑11.65)}$ |
| | | | *Image encoder tuning for* CLIP-ViT-B/16 | | | |
| Pseudo-label (Lee et al., 2013) | 69.11 | 62.15 | 63.56 | 80.03 | 49.45 | 63.80 |
| TPT (Manli et al., 2022) | 69.42 | 61.62 | 63.70 | 79.74 | 49.47 | 63.63 |
| KD (Hinton et al., 2015) | 70.92 | 66.39 | 65.01 | 82.12 | 53.51 | 66.76 |
| ATKD (Guo et al., 2020) | 70.51 | 70.66 | 65.54 | 85.12 | 53.56 | 68.72 |
| **RLCF** | 74.85$_{(↑8.12)}$ | 73.71$_{(↑25.84)}$ | 69.77$_{(↑8.91)}$ | 86.19$_{(↑12.21)}$ | 57.10$_{(↑11.01)}$ | 71.69$_{(↑14.49)}$ |
| **RLCF-S** | 75.34$_{(↑8.61)}$ | 75.00$_{(↑27.13)}$ | 70.08$_{(↑9.22)}$ | 86.97$_{(↑12.99)}$ | 57.75$_{(↑11.66)}$ | 72.45$_{(↑15.25)}$ |
| **RLCF-S-M** | 75.48$_{(↑8.75)}$ | 75.16$_{(↑27.29)}$ | 70.42$_{(↑9.56)}$ | 87.23$_{(↑13.25)}$ | 57.73$_{(↑11.64)}$ | 72.64$_{(↑15.44)}$ |

(a) ECE on IN-A. lower is better.    (b) Acc. of TPT *w.r.t.* steps.    (c) Acc. of RLCF *w.r.t.* steps.

Figure 5: **ECE and average accuracy on ImageNet-A/V2/R.** Prompt tuning with CLIP-ViT-B/16.

**Baselines** We compare RLCF with few-shot prompt tuning methods for CLIP — CoOp (Zhou et al., 2021) and CoCoOp (Zhou et al., 2022) (16 shots on ImageNet), state-of-the-art test-time prompt tuning methods — TPT (Manli et al., 2022), and knowledge distillation (KD (Hinton et al., 2015), ATKD (Guo et al., 2020)), which use the reward model as the teacher during test time. TPT + CoOp means TPT adopts the learned prompts of CoOp as the initialization, otherwise, TPT uses token embedding of a hard prompt "a photo of a" as initial weights. For all prompt tuning methods, the length of learnable prompts is 4. Results of Pseudo-label (Lee et al., 2013) are also presented.

**Implementation details** For prompt tuning, the learning rate is 7e-3, the weight decay value is 5e-4, and the optimizer is AdamW (Loshchilov & Hutter, 2019). For image encoder tuning, the learning rate is decreased to 1e-5. Given a test sample, the parameters will be optimized for 3 steps to maximize the reward of the top-3 (sampling factor $K = 3$) predictions. The momentum coefficient $m = 0.9998$ and update interval $B_s = 64$ for the momentum buffer.

**Results** In Table 1, RLCF largely improves the zero-shot generalization capacity of CLIP-ViT-B/16 and outperforms previous methods. Notably, on ImageNet-A/V2/R, *RLCF with CLIP-ViT-B/16 surpasses the reward model — CLIP-ViT-L/14*. This shows that RLCF effectively combines the capabilities of both the TTA model and the reward model through the feedback mechanism, something that KD or pseudo-label cannot achieve. RLCF significantly outperforms the entropy minimization method — TPT. TPT can only learn from the TTA model itself and lacks awareness of the correctness of its predictions. Figure 5a presents the expected calibration error (ECE) (Guo et al., 2017) of TPT and RLCF. The ECE of the two both increases along with the TTA steps, but the ECE of RLCF is clearly lower. This means the output of RLCF better reflects its uncertainty about the input and is more reliable. In Figure 1a&8, RLCF provides multiple positive scores for various objects, preventing the model from becoming blindly confident. In Figure 5b, the top-5 accuracy of TPT drops with more steps. The model is stuck in its incorrect predictions and pushes away the ground truth as shown in Figure 1a. By contrast, there is no such issue for RLCF in Figure 5c.

Ablation study about sampling factors and reward model choices can be found in Appendix B.

Table 2: **TTA for zero-shot text-image retrieval**. KD uses CLIP-ViT-L/14 as the teacher model. Improvement in Recall@1 with RLCF compared to the CLIP-ViT-B/16 baseline is in (↑blue).

| Method | MS-COCO (5K test images) | | | | | | Flickr30K (1K test images) | | | | | |
| | text-to-image | | | image-to-text | | | text-to-image | | | image-to-text | | |
| | R@1 | R@5 | R@10 | R@1 | R@5 | R@10 | R@1 | R@5 | R@10 | R@1 | R@5 | R@10 |
|---|---|---|---|---|---|---|---|---|---|---|---|---|
| | *Zero-shot baseline* | | | | | | | | | | | |
| CLIP-ViT-B/16 | 33.0 | 58.2 | 68.9 | 52.5 | 76.8 | 84.6 | 62.2 | 85.7 | 91.8 | 81.2 | 96.4 | 98.5 |
| CLIP-ViT-L/14 | 36.1 | 60.9 | 71.1 | 56.2 | 78.9 | 86.9 | 64.6 | 87.1 | 92.1 | 85.3 | 97.2 | 99.1 |
| CLIP-ViT-L/14-336 | 36.6 | 60.9 | 71.0 | 57.3 | 80.6 | 87.8 | 67.1 | 88.9 | 93.2 | 86.6 | 98.0 | 99.1 |
| | *TTA for CLIP-ViT-B/16* | | | | | | | | | | | |
| Pseudo-label (Lee et al.) | 33.0 | 57.8 | 68.2 | 52.4 | 72.4 | 81.8 | 62.2 | 85.3 | 91.7 | 81.1 | 93.2 | 97.8 |
| KD (Hinton et al.) (steps: 3) | 37.6 | 61.0 | 70.7 | 57.0 | 79.0 | 86.3 | 66.9 | 87.9 | 92.9 | 85.3 | 97.5 | 98.5 |
| KD (Hinton et al.) (steps: 5) | 34.6 | 59.7 | 69.8 | 53.7 | 76.6 | 84.5 | 61.4 | 86.0 | 91.7 | 83.1 | 95.9 | 97.9 |
| **RLCF** | 37.3$_{(↑4.3)}$ | 62.7 | 71.5 | 59.1$_{(↑6.6)}$ | 80.1 | 86.9 | 67.1$_{(↑4.9)}$ | 89.1 | 93.2 | 87.3$_{(↑6.1)}$ | 97.2 | 98.8 |
| **RLCF-S** | 38.3$_{(↑5.3)}$ | 63.4 | 72.5 | 60.8$_{(↑8.3)}$ | 80.8 | 87.5 | 68.5$_{(↑6.3)}$ | 90.0 | 93.7 | 88.3$_{(↑7.1)}$ | 97.7 | 98.9 |
| **RLCF-S-M** | 38.4$_{(↑5.4)}$ | 63.5 | 72.6 | 60.8$_{(↑8.3)}$ | 80.5 | 87.6 | 68.5$_{(↑6.3)}$ | 90.2 | 93.7 | 88.1$_{(↑6.9)}$ | 97.7 | 98.9 |

## 4.2 ZERO-SHOT TEXT-IMAGE RETRIEVAL

**Implementation details** For text-image retrieval, we use the test set of Flickr30K (Plummer et al., 2015) and test split of MS-COCO (Lin et al., 2014) divided by Karpathy *et al.* (Karpathy & Fei-Fei, 2015). Each image in the two test sets corresponds to 5 sentences. CLIP-ViT-B/16 is adopted as the retrieval model. The learning rate is 1e-6, the weight decay value is 5e-4, and AdamW optimizer is used. For MS-COCO, the sampling factor $K = 12$ for text-to-image retrieval, and $K = 20$ for the other case. For Flickr30K, $K = 12$ and $K = 16$ for text-to-image and image-to-text retrieval, respectively. The adaptation steps are 8. For the momentum buffer, $m = 0.9998$ and $B_s = 64$. We also compare RLCF with knowledge distillation (KD) with CLIP-ViT-L/14 as the teacher.

**Results** Table 2 presents the retrieval results on MS-COCO and Flickr30K. RLCF demonstrates significant improvement compared to the zero-shot baseline and even outperforms the most powerful CLIP-ViT-L/14-336. Similar phenomena are also observed in zero-shot classification. The feedback mechanism reserves the merits of the TTA model and makes the TTA model improve with the reward model. In contrast, KD or pseudo-label forces the student to mimic the teacher regardless of the correctness of the teacher as discussed in Sec. 3.2.2. In KD for supervised classification (Hinton et al., 2015; Zhao et al., 2022; Wang et al., 2021b), the student is generally worse than the teacher due to their capacity gap and incomplete learning. Nevertheless, RLCF can surpass the powerful reward model with the feedback mechanism during test time in a zero-shot circumstance.

## 4.3 IMAGE CAPTIONING

**Datasets** To test the adaptation ability of RLCF for captioning models in a zero-shot or cross-domain condition, we train the captioning model on MS-COCO train set (Lin et al., 2014) and test it on the test set of Flickr30K (Plummer et al., 2015) and validation set of NoCaps (Agrawal et al., 2019). NoCaps validation set contains three splits according to whether contains MS-COCO objects: in domain contains only MS-COCO objects, near domain contains both MS-COCO and novel objects, and out domain contains only novel objects.

**Implementation details** CLIPCap (Mokady et al., 2021) and CapDec (Nukrai et al., 2022), two LLM-based methods, are chosen as the captioning models. The two have the same architecture, while CLIPCap is trained with CLIP-ViT-B/16 image embedding and CapDec is trained with CLIP-ViT-B/16 text embedding. The projector in Figure 4 is an 8-layer transformer encoder that contains about 43M parameters. The LLM is an OPT-125M (Zhang et al., 2022b). During TTA, we only tune the parameters of the projector. For CLIPCap, the learning rate is 2e-6, and sampling factor $K = 10$; for CapDec, the learning rate is 5e-6 on Flickr30K, 3e-6 on NoCaps, and $K = 6$. No weight decay is applied. The optimizer is AdamW. The TTA step is 4. After TTA, captions are generated with a beam search with a width of 5 and the final caption is the one with the highest score.

**Results** Table 3 presents results for image captioning. A weakly supervised method — MAGIC (Su et al., 2022) and a zero-shot method —DeCap (Li et al., 2023), are included for reference. The reported metrics include BLEU@4, CIDEr, SPICE, and RefCLIPScore (Hessel et al., 2021). Ref-CLIPScore reflects the similarity between generated text and reference captions. The improvements in CIDEr metric (Vedantam et al., 2015) are highlighted. For all metrics, both CapDec and CLIPCap with RLCF significantly improve upon the baselines. This demonstrates the strong generalization ability of RLCF in image captioning, even with a single test sample. It is noteworthy that CLIPCap

Table 3: **TTA for image captioning**. B@4 for BLEU@4, C for CIDEr, S for SPICE, and Ref-C for RefCLIPScore. The gain of well-recognized CIDEr metric is in (↑blue).

| Method | MS-COCO ⟹ NoCaps | | | | | | | | | MS-COCO ⟹ Flickr30K Karpathy's test split | | | |
| | in domain | | | near domain | | | out domain | | | | | | |
| | B@4 | C | S | B@4 | C | S | B@4 | C | S | B@4 | C | S | Ref-C |
|---|---|---|---|---|---|---|---|---|---|---|---|---|---|
| MAGIC (Su et al., 2022) | - | - | - | - | - | - | - | - | - | 5.2 | 18.3 | 5.7 | - |
| DeCap (Li et al., 2023) | - | 72.7 | - | - | 61.9 | - | - | 43.9 | - | 17.7 | 42.0 | 13.8 | - |
| *TTA for CapDec (**zero-shot**)* | | | | | | | | | | | | | |
| CapDec (Nukrai et al., 2022) | 32.4 | 62.6 | 10.3 | 29.2 | 54.0 | 9.6 | 17.2 | 31.7 | 6.4 | 19.3 | 37.0 | 11.7 | 74.1 |
| + RLCF | 33.3 | 68.0$_{(↑5.3)}$ | 10.7 | 30.3 | 57.9$_{(↑3.9)}$ | 10.3 | 17.6 | 35.5$_{(↑3.8)}$ | 6.9 | 20.3 | 41.9$_{(↑4.9)}$ | 12.7 | 75.7 |
| + RLCF-S | 34.0 | 68.3$_{(↑5.7)}$ | 10.8 | 30.3 | 58.5$_{(↑4.5)}$ | 10.3 | 17.7 | 35.3$_{(↑3.6)}$ | 6.9 | 20.1 | 41.6$_{(↑4.6)}$ | 12.7 | 75.7 |
| *TTA for CLIPCap (**cross-domain**)* | | | | | | | | | | | | | |
| CLIPCap (Mokady et al., 2021) | 36.3 | 76.9 | 11.9 | 34.8 | 73.5 | 11.0 | 22.5 | 54.6 | 8.6 | 21.8 | 49.3 | 13.1 | 76.7 |
| + RLCF | 38.6 | 84.0$_{(↑7.1)}$ | 12.5 | 36.1 | 79.6$_{(↑6.1)}$ | 11.8 | 24.7 | 63.8$_{(↑9.2)}$ | 9.6 | 23.3 | 56.6$_{(↑7.3)}$ | 14.5 | 79.4 |
| + RLCF-S | 38.7 | 84.7$_{(↑7.8)}$ | 12.6 | 35.8 | 79.7$_{(↑6.2)}$ | 11.8 | 24.2 | 63.1$_{(↑8.5)}$ | 9.5 | 23.6 | 57.8$_{(↑8.5)}$ | 14.6 | 79.4 |

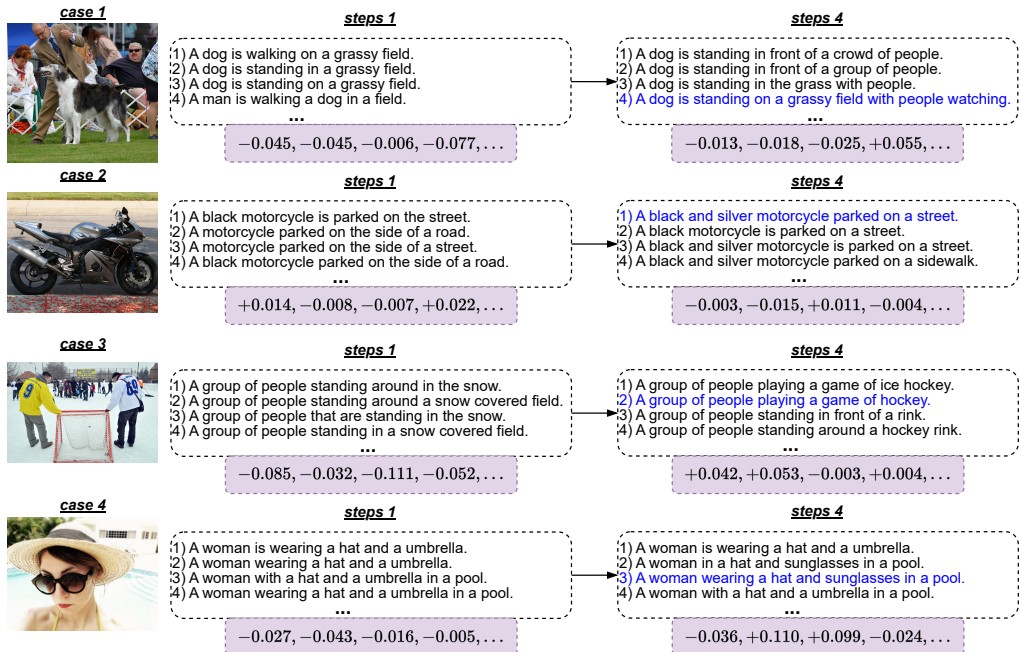

Figure 6: **Intermediate generated captions of CLIPCap and the CLIP reward**. The sampling factor $K = 10$, only 4 candidates are shown here. The final generated caption is in blue.

achieves greater improvements in CIDEr (up to 9.2) compared to CapDec. CLIPCap can also use a large sampling factor $K$. This is possible because CLIPCap can generate higher-quality candidate captions. The results of RLCF-S-M are not shown as it is no better than RLCF-S.

**Qualitative results** Figure 6 displays the intermediate-generated captions and their corresponding rewards. The visualization reveals that the CLIP reward model favors captions that provide a holistic description of the image. Through feedback, the generation of such captions is encouraged. During TTA, captions aligned with the preferences of CLIP are given higher priority. Please refer to Figure 7 in Appendix A for more visualization cases.

## 5 CONCLUSION

In this work, we introduce reinforcement learning with CLIP feedback (RLCF) to improve the zero-shot generalization ability of VLMs on the fly. A novel reward function with CLIP is developed. We instantiate three TTA pipelines for image classification, text-image retrieval, and image captioning with task-specific sampling strategies and parameter tuning manners. With RLCF, the zero-shot generalization capacity of various VLMs is boosted significantly. We hope RLCF can provide heuristic information for future research that employs TTA with feedback from large foundation models.

## ACKNOWLEDGMENTS

This work was supported in part by the Australian Research Council (ARC) under Grant DP200100938. Thanks Chao Liang for his helpful discussions.

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

APPENDIX

# A  VISUALIZATION RESULTS

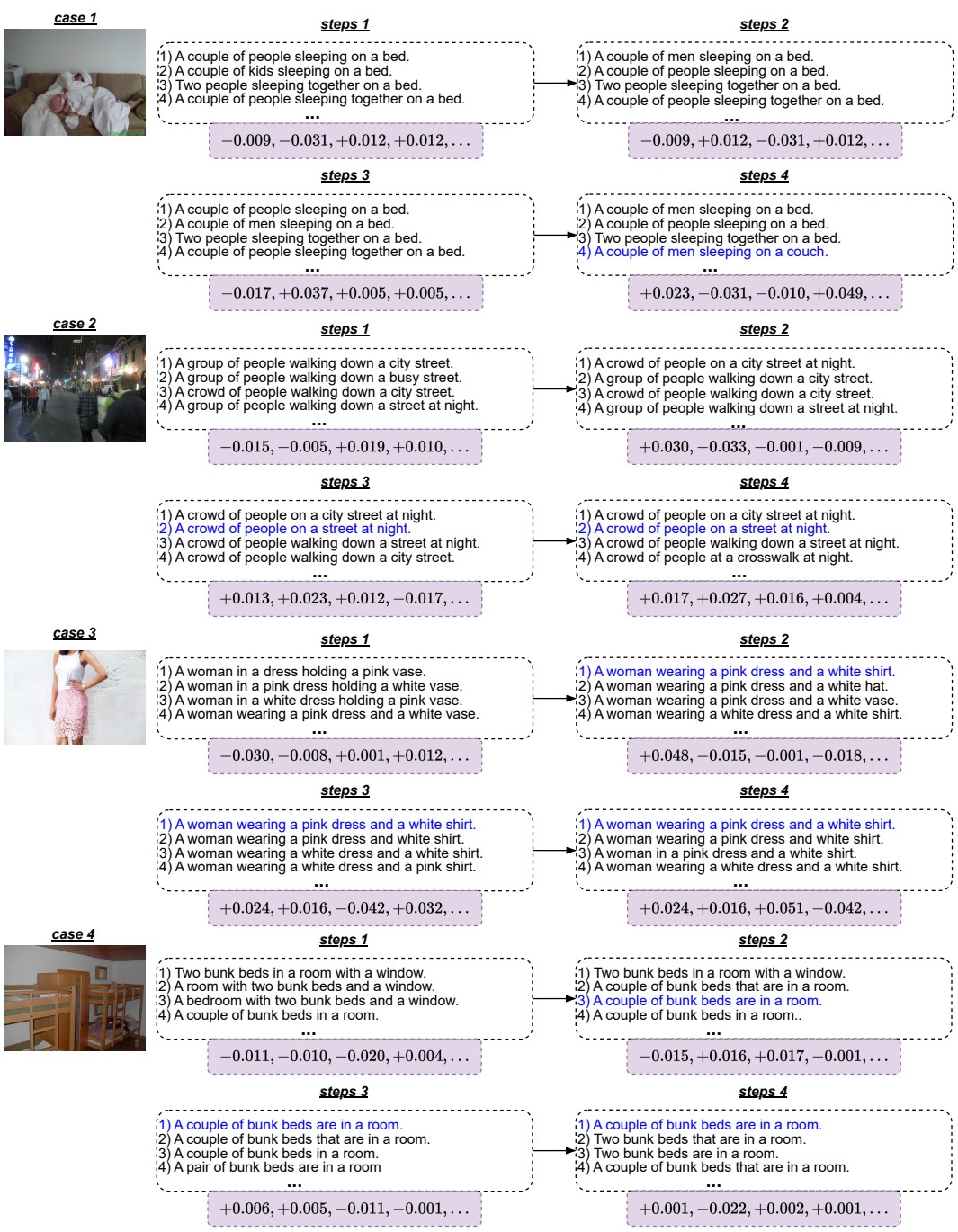

Figure 7: **Generated captions at each TTA step of CLIPCap and the CLIP reward**. The sampling factor $K = 10$, only 4 candidates are shown here. The final generated caption is in blue.

In Figure 7, we show more captioning examples like that in Figure 6. These samples illustrate how CLIP reward helps the captioning model select a description that matches the picture more closely.

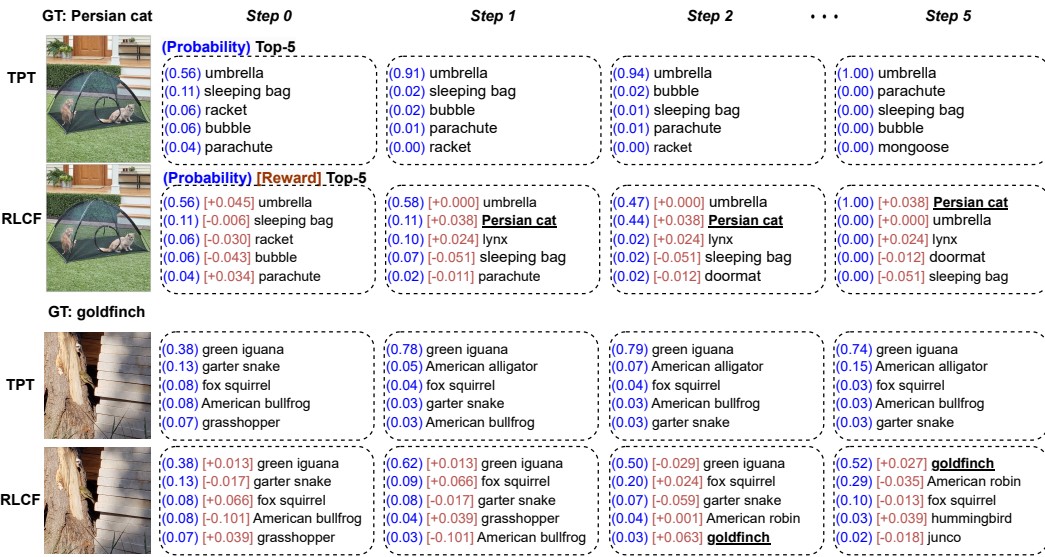

Figure 8: **Top-5 predictions, probabilities, and rewards with different TTA steps**. Prompt tuning for CLIP-ViT-B/16. For RLCF, sampling factor $K = 5$. We perform 5 TTA steps and step 0 means the original prediction. Images from ImageNet-A.

In Figure 8, we provide visualization of top-5 predictions with different TTA steps like Figure 1a. In Figure 8, the ground truth is not in the top-5 predictions. In such cases, TPT cannot find the object by minimizing the entropy of model outputs. By contrast, RLCF can discover the ground truth by pushing away the objects with negative feedback.

# B ABLATION STUDY

## B.1 SAMPLING FACTOR

When using RLCF with TTA for various VL tasks, we sample $K$ candidates from output distribution for reward calculations. In this section, we examine how $K$ affects different tasks and models.

**Image classification on OOD data** Figure 9 shows the average top-1 and top-5 accuracy on ImageNet-A, ImageNet-V2, and ImageNet-R for different $K$ in image classification. RLCF reduces to pseudo-label (Lee et al., 2013) when $K = 1$. A larger $K$ improves top-5 accuracy, but not top-1. Too many sampled classes may make the optimization process difficult for policy gradient. For example, when $K = 5$ and only one class gets a positive score and other classes get negative scores, pushing away 4 negative classes may cause unpredictable behavior and make the model miss the ground truth after gradient updating.

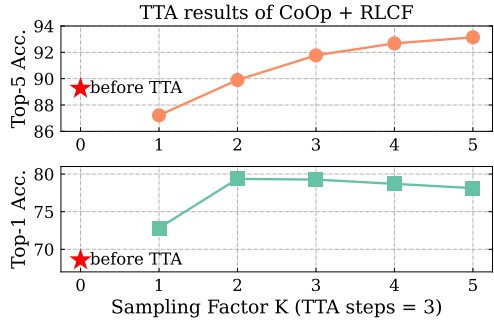

Figure 9: **Different sampling factors $K$ in image classification on OOD data**.

**Zero-shot text-image retrieval** Table 4 presents the effect of sampling factor $K$ in zero-shot text-image retrieval. Similar to image classification, a larger $K$ generally leads to better Recall@5 and Recall@10 compared to a smaller $K$. However, a smaller $K$ tends to produce better Recall@1 in

Table 4: **Different sampling factors $K$ in zero-shot text-image retrieval**. $K_{t2i}$ for sampling in text-to-image retrieval, and $K_{i2t}$ for sampling in image-to-text retrieval.

| Method | | MS-COCO (5K test images) | | | | | | Flickr30K (1K test images) | | | | | |
|---|---|---|---|---|---|---|---|---|---|---|---|---|---|
| | | text-to-image | | | image-to-text | | | text-to-image | | | image-to-text | | |
| | | R@1 | R@5 | R@10 | R@1 | R@5 | R@10 | R@1 | R@5 | R@10 | R@1 | R@5 | R@10 |
| | | | | | | | *Zero-shot baseline* | | | | | | |
| CLIP-ViT-B/16 | | 33.0 | 58.2 | 68.9 | 52.5 | 76.8 | 84.6 | 62.2 | 85.7 | 91.8 | 81.2 | 96.4 | 98.5 |
| $K_{t2i}$ | $K_{i2t}$ | | | | | *TTA for* CLIP-ViT-B/16 with **RLCF** | | | | | | | |
| 6 | 14 | 37.8 | 61.1 | 68.1 | 59.1 | 80.3 | 86.5 | 66.6 | 87.9 | 91.5 | 86.7 | 97.6 | 98.6 |
| 8 | 16 | 37.6 | 62.1 | 69.6 | 59.3 | 80.4 | 86.8 | 66.7 | 89.0 | 92.1 | 87.3 | 97.2 | 98.8 |
| 10 | 18 | 37.4 | 62.4 | 70.8 | 59.5 | 80.1 | 86.9 | 67.0 | 89.0 | 92.7 | 87.1 | 97.2 | 98.7 |
| 12 | 20 | 37.3 | 62.7 | 71.5 | 59.1 | 80.1 | 86.9 | 67.1 | 89.1 | 93.2 | 87.2 | 97.3 | 98.6 |
| 14 | 22 | 37.0 | 62.5 | 72.4 | 59.8 | 80.2 | 86.8 | 66.9 | 89.0 | 93.4 | 87.3 | 97.3 | 98.6 |
| 16 | 24 | 36.8 | 62.4 | 72.5 | 59.5 | 80.5 | 87.3 | 66.9 | 88.8 | 93.5 | 87.6 | 96.9 | 98.7 |
| 18 | 26 | 36.9 | 62.5 | 72.5 | 59.2 | 80.3 | 87.0 | 66.7 | 88.8 | 93.5 | 87.3 | 96.8 | 98.6 |

Table 5: **Different sampling factors $K$ in image captioning**. Metrics B@4 for BLEU@4, C for CIDEr, S for SPICE, and Ref-C for RefCLIPScore.

| Method | MS-COCO $\Longrightarrow$ NoCaps | | | | | | | | | MS-COCO $\Longrightarrow$ Flickr30K | | | |
|---|---|---|---|---|---|---|---|---|---|---|---|---|---|
| | in domain | | | near domain | | | out domain | | | Karpathy's test split | | | |
| | B@4 | C | S | B@4 | C | S | B@4 | C | S | B@4 | C | S | Ref-C |
| | | | | | | *TTA for* CapDec (**zero-shot**) | | | | | | | |
| CapDec (Nukrai et al., 2022) | 32.4 | 62.6 | 10.3 | 29.2 | 54.0 | 9.6 | 17.2 | 31.7 | 6.4 | 19.3 | 37.0 | 11.7 | 74.1 |
| + **RLCF** ($K = 4$) | 32.7 | 65.5 | 10.7 | 30.0 | 57.5 | 10.2 | 17.1 | 34.6 | 6.8 | 20.2 | 40.8 | 12.5 | 75.4 |
| + **RLCF** ($K = 6$) | 33.3 | 68.0 | 10.7 | 30.3 | 57.9 | 10.3 | 17.6 | 35.5 | 6.9 | 20.3 | 41.9 | 12.7 | 75.7 |
| + **RLCF** ($K = 8$) | 32.7 | 65.5 | 10.7 | 30.0 | 57.5 | 10.2 | 17.1 | 34.6 | 6.8 | 20.2 | 41.8 | 12.9 | 75.8 |
| | | | | | | *TTA for* CLIPCap (**cross-domain**) | | | | | | | |
| CLIPCap (Mokady et al., 2021) | 36.3 | 76.9 | 11.9 | 34.8 | 73.5 | 11.0 | 22.5 | 54.6 | 8.6 | 21.8 | 49.3 | 13.1 | 76.7 |
| + **RLCF** ($K = 8$) | 38.4 | 83.3 | 12.5 | 36.0 | 78.5 | 11.6 | 24.1 | 61.4 | 9.3 | 22.7 | 55.8 | 14.3 | 79.3 |
| + **RLCF** ($K = 10$) | 38.6 | 84.0 | 12.5 | 36.1 | 79.6 | 11.8 | 24.7 | 63.8 | 9.6 | 23.3 | 56.6 | 14.5 | 79.4 |
| + **RLCF** ($K = 12$) | 38.5 | 82.0 | 12.5 | 35.9 | 80.2 | 11.9 | 24.4 | 63.1 | 9.5 | 23.1 | 57.5 | 14.6 | 79.4 |

most cases. In MS-COCO and Flickr30K, one image has 5 reference captions, so the sampling factor for image-to-text retrieval is larger than $K$ for text-to-image retrieval.

**Image captioning** Table 5 illustrates the effect of different values of sampling factor $K$ in cross-domain image captioning. The optimal $K$ varies for different image captioning models. CLIPCap has better captioning capabilities than CapDec, so it can produce better candidates. Therefore, a larger $K$ is suitable for CLIPCap.

From the ablation study of sampling factor $K$, we find that the choice of $K$ depends on the tasks and VLMs. Different tasks and models require various sampling strategies.

## B.2 DIFFERENT REWARD MODELS

RLCF relies on the good quality of the CLIP reward models. In this section, we show the influence of different CLIP reward models in image classification and image captioning.

As shown in Table 6, RLCF is robust to different reward models. Compared to the baseline CoOp, RLCF can achieve improvements even with a CLIP-RN50×4 as the reward model, which is worse than the prompt tuning model CLIP-ViT-B/16. When the prompt tuning model and the reward model are the same, RLCF is also better than the state-of-the-art test-time prompt tuning method — TPT (Manli et al., 2022). With CLIP-ViT-L/14 as the reward model, RLCF with prompt tuning is slightly better than the ensemble result. It is worth noting that RLCF with image encoder tuning in Table 1 is obviously better than the ensemble results in the OOD average performance. Compared to the ensemble method, RLCF can adapt to the test distribution with the feedback mechanism. This is why RLCF shows better performance than the ensemble results.

We also test different reward models in image captioning. Results are shown in Table 7. CapDec and CLIPCap both use CLIP-ViT-B/16 as the image embedding extractor. RLCF with different reward

Table 6: **RLCF with different reward models in image classification**. "–" for not available.

| Method | Reward Model | ImageNet-A | ImageNet-V2 | ImageNet-R | ImageNet-Sketch | OOD Average |
|---|---|---|---|---|---|---|
| | | | *Zero-shot baseline* | | | |
| CLIP-RN50×4 | – | 39.66 | 58.72 | 69.10 | 42.73 | 52.55 |
| CLIP-ViT-B/16 | – | 47.87 | 60.86 | 73.98 | 46.09 | 57.20 |
| CLIP-ViT-L/14 | – | 68.82 | 67.80 | 85.40 | 57.84 | 69.97 |
| Ensemble (B/16 + L/14) | – | 65.94 | 69.02 | 85.92 | 57.98 | 69.72 |
| | | | *Prompt tuning for* CLIP-ViT-B/16 | | | |
| CoOp (Zhou et al., 2021) | – | 49.71 | 64.20 | 75.21 | 47.99 | 59.28 |
| TPT + CoOp (Manli et al.) | – | 57.95 | 66.83 | 77.27 | 49.29 | 62.84 |
| RLCF + CoOp | CLIP-RN50×4 | 52.06 | 64.62 | 76.00 | 48.42 | 60.28 |
| RLCF + CoOp | CLIP-ViT-B/16 | 61.66 | 67.04 | 78.06 | 49.70 | 64.12 |
| RLCF + CoOp | CLIP-ViT-L/14 | 69.74 | 70.62 | 84.51 | 56.49 | 70.34 |

Table 7: **RLCF with different reward models in image captioning**. Metrics M for METEOR, C for CIDEr, S for SPICE, and Ref-C for RefCLIPScore. We keep the embedding extractor for the two methods as CLIP-ViT-B/16.

| Method | Embed. Extractor | Reward Model | MS-COCO ⟹ NoCaps | | | | | | | | | | | |
| | | | in domain | | | | near domain | | | | out domain | | | |
| | | | M | C | S | Ref-C | M | C | S | Ref-C | M | C | S | Ref-C |
|---|---|---|---|---|---|---|---|---|---|---|---|---|---|---|
| | | | *TTA for* CapDec (Nukrai et al., 2022) (**zero-shot**) | | | | | | | | | | | |
| CapDec | CLIP-ViT-B/16 | – | 23.9 | 62.6 | 10.3 | 75.5 | 22.3 | 54.0 | 9.6 | 74.2 | 17.2 | 31.7 | 6.4 | 71.3 |
| + **RLCF** | CLIP-ViT-B/16 | CLIP-ViT-B/16 | 24.3 | 65.5 | 10.7 | 76.6 | 22.7 | 57.2 | 10.0 | 75.7 | 17.7 | 35.0 | 6.8 | 72.9 |
| + **RLCF** | CLIP-ViT-B/16 | CLIP-ViT-L/14 | 24.6 | 68.0 | 10.7 | 76.7 | 23.0 | 57.9 | 10.3 | 75.7 | 17.9 | 35.5 | 6.9 | 72.7 |
| | | | *TTA for* CLIPCap (Mokady et al., 2021) (**cross-domain**) | | | | | | | | | | | |
| CLIPCap | CLIP-ViT-B/16 | – | 26.4 | 76.9 | 11.9 | 77.8 | 24.8 | 73.5 | 11.0 | 77.6 | 20.3 | 54.6 | 8.6 | 75.7 |
| + **RLCF** | CLIP-ViT-B/16 | CLIP-ViT-B/16 | 26.9 | 81.1 | 12.3 | 80.1 | 25.5 | 78.1 | 11.7 | 80.1 | 21.2 | 62.0 | 9.5 | 78.5 |
| + **RLCF** | CLIP-ViT-B/16 | CLIP-ViT-L/14 | 27.2 | 84.0 | 12.5 | 80.3 | 25.7 | 79.6 | 11.8 | 80.1 | 21.5 | 63.8 | 9.6 | 78.5 |

Table 8: **Average GPU inference time per sample and GPU memory with different TTA steps**. Test on ImageNet-A and ImageNet-V2 with a single NVIDIA 40GB A100 GPU.

| CLIP-ViT-B/16 | TTA Steps | ImageNet-A | | | ImageNet-V2 | | |
| | | Acc. | Mem. (GB) | Time (s) | Acc. | Mem. (GB) | Time (s) |
|---|---|---|---|---|---|---|---|
| **Prompt Tuning** | | | | | | | |
| TPT + CoOp | 1 | 57.95 | 4.2 | 0.168 | 66.83 | 18.2 | 0.468 |
| TPT + CoOp | 3 | 60.13 | 4.2 | 0.320 | 66.76 | 18.2 | 1.05 |
| RLCF + CoOp | 1 | 63.07 | 6.2 | 0.197 | 69.59 | 19.8 | 0.486 |
| RLCF + CoOp | 3 | **69.74** | 6.2 | 0.348 | **70.62** | 19.8 | 1.08 |
| **Image encoder tuning** | | Acc. | Mem. (GB) | Time (s) | Acc. | Mem. (GB) | Time (s) |
| TPT + CoOp | 1 | 61.78 | 8.8 | 0.208 | 63.70 | 8.8 | 0.272 |
| TPT + CoOp | 3 | 62.07 | 8.8 | 0.384 | 64.02 | 8.8 | 0.512 |
| RLCF | 1 | 71.23 | 10.8 | 0.239 | 67.60 | 10.8 | 0.319 |
| RLCF | 3 | **73.71** | 10.8 | 0.415 | **69.77** | 10.8 | 0.543 |

models can always achieve significant improvements in the near and out domain data. This shows the robustness of RLCF to open domain scenarios with different CLIP reward models.

### B.3 GPU RUNTIME AND MEMORY

Efficiency is also important in TTA. We provide the GPU time and memory in Table 8.

**Compared to TPT, the inference time of RLCF increases by a constant amount for different TTA steps and datasets, *i.e.*, roughly 0.03s per sample**. For each sample, the CLIP reward model only needs to run the image encoder once. This is the source of the 0.03s increase. The text features of CLIP reward model are always the same because the class names are fixed.

ImageNet-A has 200 classes, while ImageNet-V2 has 1000 classes. For prompt tuning on ImageNet-V2, the input batch size of the CLIP text encoder is 1000, and we need to re-run the text encoder to update the text features after each TTA step. This is why prompt tuning is slower and consumes more memory than image encoder tuning. For image encoder tuning, the text features are unchanged and the image encoder only has a single image as input.

