# OpenReview forum: "Test-Time Adaptation with CLIP Reward for Zero-Shot Generalization in Vision-Language Models"
_ICLR.cc/2024/Conference — ICLR 2024 poster_

### Official Review · Reviewer_efhx · 2023-10-30

**Soundness:** 3 good
**Presentation:** 4 excellent
**Contribution:** 3 good
**Rating:** 8
**Confidence:** 4

**Summary:**

This paper solves test-time adaptation of Vision-Language-Models to improve the zero-shot generalization performances. Unlike previous works that rely on entropy of model outputs, the authors propose to leverage reward from CLIP model as feedback to adapt models. The proposed method based on reinforcement learning with CLIP feedback is evaluated on three tasks, including zero-shot image classification, zero-shot text-image retrieval, zero-shot and cross-domain image captioning, showing improved performances.

**Strengths:**

+ The paper solves an important task of test-time adaptation of vision-language models. VLMs have played a critical role in many CV and NLP tasks.  How to improve their zero-shot generalization ability, especially in the challenging task of test-time adaptation, is a worthwhile research problem.
+ The authors propose to leverage CLIP feedback for adapting models. This seems to be novel and is interesting to me. As many vision-language learning methods are built upon CLIP, the intergration of CLIP reward is natural.
+ The proposed RLCF method universally applies to different tasks, like zero-shot classification, text-image retrieval, image captioning. Hence, the idea may inspire research in many other VL tasks.
+ The method is simple yet effective. The experiments and analyses are extensive in the paper. For each task, RLCF is compared with a few  baseline methods, showing significant improvements.
+ The paper is very easy to follow with a clear motivation and method descriptions. Visualizations (e.g. fig.1) also look nice.

**Weaknesses:**

- The method relies on good quality of CLIP feedback. This may restrict its applications in tasks other than images with generic objects. For example, CLIP shows less satisfactory accuracies on fine-grained datasets like FGVC Aircraft, EuroSAT, CUB, etc. CLIPScore may be not informative in the fine-grained classification.
- The authors use CLIP-ViT-L and a few other variants for calculating reward. The commonly used backbone in VL learning is CLIP-ViT-B. It is understandable that CLIP-ViT-L leads to better text-image alignment. But discussions on CLIPScore with CLIP-ViT-B would be helpful to understand how robust RLCF is to unreliable rewards.

**Questions:**

- In Eq.(4), why need to clip CLIP score to be non-negative? What 'it encourages all model behaviors' means? In Eq.(5), any difference if not subtracting the expectation term?
- The authors mention RLCF-S adopts weighted reward sum and RLCF-S-M adds a momentum buffer. Could the authors explain more about this? Also as mentioned in the weakness, more comments on using less reliable CLIP like CLIP-RN50, CLIP-B would be appreciated.

---

> ### Author Response · Authors · 2023-11-18
> **Rebuttal - Part I**
>
> - **1.** Different choices of reward models and how robust RLCF is to unreliable rewards
>
>     To figure out how robust RLCF is to different reward models, we first test RLCF with different reward models in image classification. As shown in Table efhx-1, RLCF is robust to different reward models. Compared to the baseline CoOp, RLCF can achieve improvements even with a CLIP-RN50x4 as the reward model, which is worse than the prompt tuning model CLIP-ViT-B/16. During TTA, RLCF will first sample a few image-text pairs and the reward model will only give feedback to the sampled feedback. RLCF will not require the reward model to provide feedback for the image-text pairs not sampled. Namely, RLCF will not totally follow the output of the reward model. This reduces the risk of model collapse when the reward model is less informative than the tuning model.
>
> **Table efhx-1. Different reward models with RLCF for prompt tuning with CLIP-ViT-B/16 on OOD data.**
> | Method                              | Reward Model  | IN-A      | IN-V2     | IN-R      | IN-Sketch | OOD Avg.  |
> |-------------------------------------|---------------|-----------|-----------|-----------|-----------|-----------|
> | **Zero-shot**                       |               |           |           |           |           |           |
> | CLIP-RN50x4                         |               | 39.66     | 58.72     | 69.10     | 42.73     | 52.55     |
> | CLIP-ViT-B/16                       |               | 47.87     | 60.86     | 73.98     | 46.09     | 57.20     |
> | CLIP-ViT-L/14                       |               | 68.82     | 67.80     | 85.40     | 57.84     | 69.97     |
> |                                     |               |           |           |           |           |           |
> | **Prompt Tuning for CLIP-ViT-B/16** |               |           |           |           |           |           |
> | CoOp                                |               | 49.71     | 64.20     | 75.21     | 47.99     | 59.28     |
> | TPT + CoOp                          |               | 57.95     | 66.83     | 77.27     | 49.29     | 62.84     |
> | RLCF + CoOp                         | CLIP-RN50x4   | 52.06     | 64.62     | 76.00     | 48.42     | 60.28     |
> | RLCF + CoOp                         | CLIP-ViT-B/16 | 61.66     | 67.04     | 78.06     | 49.70     | 64.12     |
> | RLCF + CoOp                         | CLIP-ViT-L/14 | **69.74** | **70.62** | **84.51** | **56.49** | **70.34** |
>
> We also test different reward models in image captioning. Results are shown in Table efhx-2. CapDec and CLIPCap both use CLIP-ViT-B/16 as the image embedding extractor. RLCF with different reward models can always achieve significant improvements in the
> near and out domain data. **This shows the robustness of RLCF to open domain scenarios with different CLIP reward models.**
>
> **Table efhx-2. Different reward models with RLCF in Image captioning.** Train on COCO and test on NoCaps.
> |           | Reward Model  |          |          |          | COCO     | --->     | NoCaps   |          |          |         |
> |-----------|---------------|----------|----------|----------|----------|----------|----------|----------|----------|---------|
> |           |               |          | in       |          |          | near     |          |          | out      |         |
> |           |               | METEOR   | CIDEr    | SPICE    | METEOR   | CIDEr    | SPICE    | METEOR   | CIDEr    | SPICE   |
> | CapDec    |               | 23.9     | 62.6     | 10.3     | 22.3     | 54.0     | 9.6      | 17.2     | 31.7     | 6.4     |
> | **+RLCF** | CLIP-ViT-B/16 |   24.3   |   65.5   |   10.7   |   22.7   |   57.2   |   10.0   |   17.7   |   35.0   |   6.8   |
> | **+RLCF** | CLIP-ViT-L/14 | **24.6** | **68.0** | **10.7** | **23.0** | **57.9** | **10.3** | **17.9** | **35.5** | **6.9** |
> |           |               |          |          |          |          |          |          |          |          |         |
> | CLIPCap   |               | 26.4     | 76.9     | 11.9     | 24.8     | 73.5     | 11.0     | 20.3     | 54.6     | 8.6     |
> | **+RLCF** | CLIP-ViT-B/16 |   26.9   |   81.1   |   12.3   |   25.5   |   78.1   |   11.7   |   21.2   |   62.0   |   9.5   |
> | **+RLCF** | CLIP-ViT-L/14 | **27.2** | **84.0** | **12.5** | **25.7** | **79.6** | **11.8** | **21.5** | **63.8** | **9.6** |
>
> We add these contents to Section B.2 DIFFERENT REWARD MODELS in the appendix of the revised paper.

---

> > ### Author Response · Authors · 2023-11-18
> > **Rebuttal - Part II**
> >
> > - **2.** CLIP reward on fine-grained datasets
> >
> >     Yes, RLCF relies on the good quality of CLIP feedback. The motivation for using CLIP as the feedback source is just CLIP shows great generalization across different VL tasks.
> >
> >     Following TPT, we evaluate RLCF on 10 fine-grained classification datasets. Table efhx-3 shows the results. RLCF is still better than previous baselines at most time. However, we also notice that RLCF does not perform well on EuroSAT as RLCF works on ImageNet series datasets. EuroSAT contains satellite images, which are quite different from images with generic objects. This verifies the importance of informative and reliable rewards, and CLIP feedback may not be reliable all the time.
> >
> > **Table efhx-3. Validation on fine-grained classification datasets.**
> > | CLIP-ViT-B/16 | Flower102 | DTD       | Pets      | Cars      | UCF101    | Caltech101 | Food101   | SUN397    | Aircraft  | EuroSAT   |
> > |---------------|-----------|-----------|-----------|-----------|-----------|------------|-----------|-----------|-----------|-----------|
> > | Prompt Tuning |           |           |           |           |           |            |           |           |           |           |
> > | CoOp          | 68.71     | 41.92     | 89.14     | 64.51     | 66.55     | 93.70      | 85.30     | 64.15     | 18.47     | **46.39** |
> > | TPT           | 68.98     | 47.75     | 87.79     | 66.87     | 68.04     | 94.16      | 84.67     | 65.5      | 24.78     | 42.44     |
> > | RLCF          | **74.46** | **51.95** | **91.58** | **75.30** | **73.25** | **95.62**  | **88.00** | **69.97** | **26.31** | 42.62     |
> >
> > - **3.** Why CLIPScore need to be non-negative in Eq(4), etc
> >
> >     - **3.1** Why CLIPScore need to be non-negative in Eq(4)
> >
> >     Eq(4) is the definition of CLIPScore in [1]. [1] propose to use similarity scores of CLIP features as the evaluation metric for image captioning tasks. For a metric, [1] requires CLIPScore to be non-negative. We conjecture that this is to determine the minimum and maximum value of CLIPScore for a better comparison during evaluation.
> >
> >     Actually, we found the value of CLIPScore is larger than 0 by a large margin at most times in practice.
> >
> >
> >     - **3.2** What 'it encourages all model behaviors' means? Any difference if not subtracting the expectation term?
> >
> >     We would like to answer the two questions with a specific example. In Figure 1(b) in the paper, the first image (with sheeps in it) and three sentences have CLIP reward [+0.25, -0.09, -0.16]. This is the result with the average baseline. Without the baseline, the CLIPScore is [0.61, 0.27, 0.21]. If we do not subtract the baseline, as Eq.(3) shows, the reward model will encourage all output of the model, namely, "it encourages all model behaviors". However, some output of the TTA model maybe totally irrelevant to the input. For example, in Figure 8 in the appendix, all sampled text is irrelevant to the input. If the reward model still provides positive feedback for these irrelevant outputs, the TTA model will tend to output these irrelevant text. This may lead to unpredictable results.
> >
> >     To give a straightforward comparison, we provide the results of RLCF with and without the average baseline in Table efhx-3. Without the baseline, the TTA model collapsed. It is even worse than the zero-shot baseline. This demonstrates the necessity and effectiveness of the average baseline.
> >
> > **Table efhx-4. RLCF with and without average baseline in image classification.**
> > | CLIP-ViT-B/16     | Reward Baseline | IN-A      | IN-V2     | IN-R      |
> > |-------------------|----------|-----------|-----------|-----------|
> > | **Prompt Tuning** |          |           |           |           |
> > | RLCF              | No       |   15.28   |   19.92   |   24.82   |
> > | RLCF              | Yes      | **69.74** | **70.62** | **84.51** |
> >
> > - **4.** More details about RLCF-S and RLCF-S-M
> >
> >     The details of RLCF-S can be referred to Section 3.2.3 TEST-TIME ADAPTATION TRICKS Paragraph Multiple reward models with weights. We use ensemble feedback of three CLIP models: {CLIP-ViT-L/14-336: 10, CLIP-ViT-L/14: 5, CLIP-RN50×64: 3}. They are given a human preference score. During ensemble, these scores are normalized to 1 and used as the weighted score.
> >
> >     The details of RLCF-S-M can be referred to Section 3.2.3 TEST-TIME ADAPTATION TRICKS Paragraph Momentum buffer. A copy of the learnable parameters of the model is maintained as the momentum buffer. Given a test sample, the updated parameters are saved to the buffer with momentum.
> >
> >     We will also open-source the code in the future to help better understand the implementation details.
> >
> > [1]  Hessel, Jack and Holtzman, Ari and Forbes, Maxwell and Bras, Ronan Le and Choi, Yejin. CLIPScore: A Reference-free Evaluation Metric for Image Captioning. EMNLP 2021.

---

> > > ### Comment · Reviewer_efhx · 2023-11-22
> > > **response to rebuttal**
> > >
> > > Thanks for the authors' rebuttal. The explaination and results are reasonable to me, thus I maintain my original score.

---

> > > > ### Author Response · Authors · 2023-11-23
> > > > **response**
> > > >
> > > > Thanks for your time and response~

---

### Official Review · Reviewer_uHKV · 2023-10-30

**Soundness:** 2 fair
**Presentation:** 3 good
**Contribution:** 2 fair
**Rating:** 6
**Confidence:** 4

**Summary:**

This paper studies the test time adaptation methods for vision-language models. The authors propose a framework, called reinforcement learning with CLIP feedback  (RLCF).  Specifically, given a test sample, the VLM is optimized to maximize the CLIP reward, which is provided by a CLIP model.  RLCF can be applied to image classification, text-to-image/image-to-text retrieval, and image captioning.  Extensive experiments demonstrate the effectiveness of RLCF.

**Strengths:**

- This paper is well-organized and easy to follow.

- The proposed RLCF framework is universal and applicable across various VL tasks.

- The experiments on three tasks show the superior effectiveness of RLCF, compared to TPT, KD, and Pseudo-label.

**Weaknesses:**

- The proposed RLCF utilizes the CLIP model to provide a reward score, which is very similar to a recent work [1]. The two tweaks are sampling strategies and adding a baseline to the reward function. It weakens the novelty of the proposed method. Besides, the comparison between the proposed RLCH and [1] is missing.

- The authors state that compared to pseudo-label and KD, the feedback mechanism combines the merits of both the student and teacher. However, the authors only investigate the original version of KD proposed in 2015. Moreover, I recommend that the authors evaluate the ensemble of the student and teacher model, which can directly combine their merits.

- For the experiments on image captioning, I suggest that the authors add the results of  CLIPCap and CapDec with the CLIP-ViT-L/14 architecture (the teacher model) for a more comprehensive comparison.

- Could the authors provide more implementation details of TPT+KD?

- Typo: kD->KD on page 8


[1] Cho, Jaemin, et al. Fine-grained Image Captioning with CLIP Reward. In Findings of NAACL, 2022.

**Questions:**

- More advanced KD and the ensemble of the student and teacher model.
- The results of  CLIPCap and  CapDec with the CLIP-ViT-L/14 architecture.
- More implementation details of TPT+KD.

---

> ### Author Response · Authors · 2023-11-18
> **Rebuttal - Part I**
>
> - **1.** Different between RLCF and the previous work [1]
>
>     - In [1], CLIPScore works with grammar regularization or CIDEr metric during training, while RLCF works with independent CLIP reward function at test time. The training of the grammar head and calculation of the CIDEr metric in [1] both need reference text data, which is unavailable at test time. When CLIPScore is used as the independent reward function in [1] (its Table 2), 11.2 CIDEr is achieved, while the baseline has 110.3 CIDEr. [1] actually demonstrate that CLIP cannot be used as the sole reward function. This is different from our findings that CLIP reward can also be used as an independent reward function at test time.
>     - The difference of the problems is non-negligible. [1] adopt CLIPScore together with grammar regularization or CIDEr metric at the training stage, where batched image-text pairs are available. Thus no sampling strategies are required. By contrast, RLCF works at the test time, where only a single test sample is available. Therefore, task-specific sampling strategies are necessary when using CLIP as the reward model. [1] cannot be directly transferred to test time due to the lack of reference text data, this is also why we do not provide a direct comparison with [1].
>     - We contribute the novel reward baseline for CLIP reward as well as task-specific TTA pipelines along with sampling strategies. These contributions are important for CLIP reward to work as the independent reward function across different tasks at test time. We also demonstrate the effectiveness of CLIP reward across different tasks. These contributions are non-trivial to the community.
>
>     We add more discussion with [1] at the Section RELATED WORK in the revised paper.
>
>
>
> - **2.** More advanced KD and the ensemble of the student and teacher model.
>
>     We provide the ensemble results and two more advanced KD results (DKD[2] and ATKD[3]). The learning rates and TTA steps of the two methods are set the same asn RLCF. We set the hyperparameters of the two methods as the recommendations of the original papers.
>
>     The results are shown in **Tabel uHKV-1**. RLCF is also better than the two KD methods. It is worth noting that RLCF with CLIP-ViT-B/16 is even better than the teacher CLIP-ViT-L/14 on ImageNet-A/ImageNet-V2/ImageNet-R. By contrast, the performance of the student models with KD methods is generally worse than the teacher models (This is a general belief in KD for the image classification problem).
>
>     RLCF is also better than the ensemble results of CLIP-ViT-B/16+CLIP-ViT-L/14 by ~2% on average on the 4 OOD datasets. Compared to the ensemble method, RLCF can adapt to the test distribution with the feedback mechanism. This is why RLCF shows better performance than the ensemble results.
>
> **Tabel uHKV-1 More advanced KD and ensemble results on OOD data**
> |                        | IN-A      | IN-V2     | IN-R        | IN-Sketch  | OOD Avg.  |
> |------------------------|-----------|-----------|-------------|------------|-----------|
> | **_zero-shot_**        |           |           |             |            |           |
> | CLIP-ViT-B/16          | 47.87     | 60.86     | 73.98       | 46.09      | 57.20     |
> | CLIP-ViT-L/14          | 68.82     | 67.80     | 85.40       | 57.84      | 69.97     |
> | Ensemble (B/16 + L/14) | 65.94     | 69.02     | 85.92       | **57.98**  | 69.72     |
> |                        |           |           |             |            |           |
> | **CLIP-ViT-B/16**      |           | **Image** | **Encoder** | **Tuning** |           |
> | DKD [2]                | 67.95     | 65.28     | 82.70       | 53.65      | 67.40     |
> | ATKD [3]               | 70.66     | 65.54     | 85.12       | 53.56      | 68.72     |
> | RLCF                   | **73.71** | **69.77** | **86.19**   | 57.10      | **71.69** |
>
>   We add ATKD in Table 1 and ensemble results in Table 7 in the revised paper.
>
> [1] Cho, Jaemin, et al. Fine-grained Image Captioning with CLIP Reward. In Findings of NAACL, 2022.
>
> [2] Borui Zhao, Quan Cui, Renjie Song, Yiyu Qiu, Jiajun Liang.Decoupled Knowledge Distillation. CVPR 2022.
>
> [3] Guo, Jia, et al. "Reducing the teacher-student gap via spherical knowledge disitllation." arXiv preprint arXiv:2010.07485 (2020).

---

> ### Author Response · Authors · 2023-11-18
> **Rebuttal - Part II**
>
> - **3.** CLIPCap and CapDec with the CLIP-ViT-L/14 architecture
>
>     We show the image captioning results with CLIP-ViT-L/14 in Tabel uHKV-2. With CLIP-ViT-L/14 as the image embedding extractor, RLCF with CLIP-ViT-L/14 as the reward model still achieves significant improvements. Specifically, RLCF+CLIPCap achieve more than +6% improvement in the CIDEr metric on the out domain data of NoCaps.
>
>     With different configurations of image embedding extractor and CLIP reward model, RLCF can always achieve substantial improvements in the near domain and out domain data. This shows that RLCF is robust to the domain shift. It is desirable for the TTA problem.
>
> **Tabel uHKV-2 CapDec and CLIPCap with CLIP-ViT-L/14 architecture on NoCaps** (train on COCO image or text, test on NoCaps). The reward model is always CLIP-ViT-L/14.
> |           | CLIP Projector |          |          |          | COCO     |  --->    | NoCaps   |          |          |         |
> |-----------|----------------|----------|----------|----------|----------|----------|----------|----------|----------|---------|
> |           |                |          | in       |          |          | near     |          |          | out      |         |
> |           |                |  METEOR  | CIDEr    | SPICE    |  METEOR  | CIDEr    | SPICE    |  METEOR  | CIDEr    | SPICE   |
> | CapDec    | CLIP-ViT-B/16  |   23.9   | 62.6     | 10.3     |   22.3   | 54.0     | 9.6      |   17.2   | 31.7     | 6.4     |
> | **+RLCF** | CLIP-ViT-B/16  | **24.6** | **68.0** | **10.7** | **23.0** | **57.9** | **10.3** | **17.9** | **35.5** | **6.9** |
> |           |                |          |          |          |          |          |          |          |          |         |
> | CapDec    | CLIP-ViT-L/14  |   24.3   |   65.5   |   10.5   |   22.7   |   57.4   |    9.9   |   17.5   |   32.8   |   6.3   |
> | **+RLCF** | CLIP-ViT-L/14  | **24.9** | **66.7** | **11.1** | **23.2** | **60.6** | **10.5** | **18.3** | **36.8** | **6.8** |
> |           |                |          |          |          |          |          |          |          |          |         |
> | CLIPCap   | CLIP-ViT-B/16  |   26.4   | 76.9     | 11.9     |   24.8   | 73.5     | 11.0     |   20.3   | 54.6     | 8.6     |
> | **+RLCF** | CLIP-ViT-B/16  | **27.2** | **84.0** | **12.5** | **25.7** | **79.6** | **11.8** | **21.5** | **63.8** | **9.6** |
> |           |                |          |          |          |          |          |          |          |          |         |
> | CLIPCap   | CLIP-ViT-L/14  |   26.2   |   77.9   |   11.4   |   25.2   |   76.7   |   11.3   |   20.7   |   58.4   |   8.9   |
> | **+RLCF** | CLIP-ViT-L/14  | **27.1** | **80.5** | **12.4** | **26.0** | **81.1** | **12.1** | **21.6** | **65.2** | **9.8** |
>
> We also add CapDec and CLIPCap with CLIP-ViT-B/16 as the reward model in Table 7 in the revised paper.
>
>
> - **4.** More implementation details of TPT+KD
>
>   For test time prompt tuning with TPT+KD in image classification, we inherit the prompt weights of CoOp. During TTA, we minimize the cross entropy between logits from the teacher and student model for the samples after confidence selection. The teacher model is always CLIP-ViT-L/14.
>
>   As for the learn hyper-parameters, the learning rate and TTA steps are the same as RLCF, i.e., 7e-3 and 3 TTA steps. More TTA steps will not benefit TPT+KD as shown in the retrieval task in Table 2.
>
> [1] Cho, Jaemin, et al. Fine-grained Image Captioning with CLIP Reward. In Findings of NAACL, 2022.
>
> [2] Borui Zhao, Quan Cui, Renjie Song, Yiyu Qiu, Jiajun Liang.Decoupled Knowledge Distillation. CVPR 2022.
>
> [3] Guo, Jia, et al. "Reducing the teacher-student gap via spherical knowledge disitllation." arXiv preprint arXiv:2010.07485 (2020).

---

> > ### Comment · Reviewer_uHKV · 2023-11-23
> >
> > Thank you for your response. It helps clear up my concerns. I have thus decided to increase my score to 6.

---

> > > ### Author Response · Authors · 2023-11-23
> > > **Response**
> > >
> > > Thanks for your time and informative review~

---

### Official Review · Reviewer_9Cat · 2023-11-02

**Soundness:** 3 good
**Presentation:** 3 good
**Contribution:** 3 good
**Rating:** 6
**Confidence:** 5

**Summary:**

This article suggests a Reinforcement Learning strategy using a CLIP-based model suitable for implementation during test time in the zero-shot learning framework. The suggested concept was tested on three distinct tasks within the zero-shot framework: image classification, image retrieval, and image captioning.

**Strengths:**

1- The paper introduces a novel reinforcement learning-based reward function to enhance the efficacy of the CLIP-based approach, which can be employed during test time for a new dataset.

2- The suggested concept is applied to three distinct tasks: image classification, test to image retrieval, and image captioning, all of which are experimented within the zero-shot framework.

3-  The experiments were conducted on the Imagenet dataset for all three tasks - image classification, image retrieval, and image captioning, demonstrating superior performance compared to recent state-of-the-art methodologies.

**Weaknesses:**

1- The proposed method is employed within the zero-shot learning framework for all three tasks: image classification, image retrieval, and image captioning. It would indeed be intriguing to explore whether it can also be applied within a few-shot learning framework?

2-  Figure-2 illustrates the application of a data augmentation strategy for image classification. Is this same strategy for augmentation also utilized for textual data?

3-  Figure-6 demonstrates the visual outcomes of step-1 and step-4, depicting the progression from the worst to the best. Including results from all steps, such as step-1, step-2, step-3, and step-4, would offer a more comprehensive understanding of how the transformation from the worst results to the best unfolds.

**Questions:**

Please see the questions raised in the weaknesses section.

---

> ### Author Response · Authors · 2023-11-18
> **Rebuttal**
>
> - **1.** Can RLCF be applied within a few-shot learning framework.
>
>     Actually, in Table 1 for image classification, for prompt tuning with CLIP-ViT-B-16, RLCF inherits the prompt weights from CoOp[1]. CoOp is a few-shot learning method (16 shots on ImageNet). RLCF works well with weights from CoOp and surpasses the CoOp baseline by a large margin. This case demonstrates that RLCF can also work well with a few-shot method.
>
> We modify Table 1 in the revised paper to make this point clearer. Specifically, we add the test prompt tuning without CoOp pre-trained prompt weights and highlight the usage of CoOp weights (In the original submission version, we mention the usage of CoOp in the implementation details). These contents are also shown in Table 9Cat-1.
>
> **Table 9Cat-1. RLCF with and without CoOp.**
> | CLIP-ViT-B/16     | ImageNet  | IN-A      | IN-V2     | IN-R      | IN-Sketch |
> |-------------------|-----------|-----------|-----------|-----------|-----------|
> | **Prompt tuning** |           |           |           |           |           |
> |        CoOp       |   71.51   |   49.71   |   64.20   |   75.21   |   47.99   |
> | TPT               | 68.98     | 54.77     | 63.45     | 77.06     | 47.94     |
> | TPT + CoOp        | 73.61     | 57.95     | 66.83     | 77.27     | 49.29     |
> | RLCF              |   73.23   | 65.45     | 67.77     | 83.35     | 54.74     |
> | RLCF + CoOp       | **76.05** | **69.74** | **70.62** | **84.51** | **56.49** |
>
>
> - **2.** Data augmentation for textual data
>
> We apply RLCF in image classification, text-image retrieval, and image captioning. Different datasets including ImageNet/MS COCO/Flickr30K/NoCaps are used. For all experiments, **no augmentations are applied for the textual data during TTA**.
>
> - **3.** All steps in Figure 6
>
>   Thanks for your suggestions. We add a new Figure 7 in the appendix of the revised paper to show the results at all TTA steps.
>
>   We display the top-4 generated captions with their CLIP reward as each TTA steps. These  samples illustrate how CLIP reward helps the captioning model select a description that matches the picture more closely
>
> [1] Kaiyang Zhou, Jingkang Yang, Chen Change Loy, and Ziwei Liu. Learning to prompt for vision language models. IJCV, 2022.

---

### Author Response · Authors · 2023-11-18
**Changes of the rebuttal version**

We appreciate the reviewers' time and efforts in reviewing our paper and providing informative feedback. Your constructive reviews have definitely improved the quality of the paper.

Here we list the changes to the paper. They are also highlighted in the revised version.
- Add more discussions with previous works in Section RELATED WORK
- Add results of more advanced KD methods and RLCF without CoOp weights in Table 1
- Add a new Figure 7 in the appendix to show all intermediate generated captions
- Add a new section B.2 DIFFERENT REWARD MODELS to discuss the influence of using different reward models in image classification and image captioning.
- Add new Table 6 and Table 7 to show the experiment results of different reward models in image classification and image captioning.

---

### Meta-Review · Area_Chair_cpzY · 2023-12-13

**Metareview:**

This paper presents a new approach to test-time adaptation for vision-language models that addresses model confidence under incorrect predictions. Through extensive experimentation across different tasks (classification, retrieval and captioning) the usefulness and performance of the method is highlighted. This work focusses on VLMs with a dual encoder setup where modality fusion happens at the output level and for future work it would be interesting to explore other variants of VLMs, e.g. having cross-modal architecture components.

**Justification For Why Not Higher Score:**

Work explores only a single type of vision-language models

**Justification For Why Not Lower Score:**

Addressing important problem of test-time adaptation in VLMs supported with extensive experimental results

---

### Decision · Program_Chairs · 2024-01-16

Accept (poster)